# A Systematic Review and Developmental Perspective on Origin of CMS Genes in Crops

**DOI:** 10.3390/ijms25158372

**Published:** 2024-07-31

**Authors:** Xuemei Zhang, Zhengpin Ding, Hongbo Lou, Rui Han, Cunqiang Ma, Shengchao Yang

**Affiliations:** 1State Key Laboratory of Conservation and Utilization of Bio-Resources in Yunnan, The Key Laboratory of Medicinal Plant Biology of Yunnan Province, Yunnan Agricultural University, Kunming 650201, China; zhangxuemeirj@163.com; 2College of Agronomy and Biotechnology, Yunnan Agricultural University, Kunming 650201, China; dzp7071@163.com (Z.D.); hongbo_lou123@163.com (H.L.); 3College of Horticulture, Nanjing Agricultural University, Nanjing 210095, China; 2020204038@stu.njau.edu.cn

**Keywords:** distant hybridization, chimeric mitochondrial gene, original CMS source, mtDNA recombination, seed germination, mitochondrial fusion and fission

## Abstract

Cytoplasmic male sterility (CMS) arises from the incompatibility between the nucleus and cytoplasm as typical representatives of the chimeric structures in the mitochondrial genome (mitogenome), which has been extensively applied for hybrid seed production in various crops. The frequent occurrence of chimeric mitochondrial genes leading to CMS is consistent with the mitochondrial DNA (mtDNA) evolution. The sequence conservation resulting from faithfully maternal inheritance and the chimeric structure caused by frequent sequence recombination have been defined as two major features of the mitogenome. However, when and how these chimeric mitochondrial genes appear in the context of the highly conserved reproduction of mitochondria is an enigma. This review, therefore, presents the critical view of the research on CMS in plants to elucidate the mechanisms of this phenomenon. Generally, distant hybridization is the main mechanism to generate an original CMS source in natural populations and in breeding. Mitochondria and mitogenomes show pleomorphic and dynamic changes at key stages of the life cycle. The promitochondria in dry seeds develop into fully functioning mitochondria during seed imbibition, followed by massive mitochondria or mitogenome fusion and fission in the germination stage along with changes in the mtDNA structure and quantity. The mitogenome stability is controlled by nuclear loci, such as the nuclear gene *Msh1*. Its suppression leads to the rearrangement of mtDNA and the production of heritable CMS genes. An abundant recombination of mtDNA is also often found in distant hybrids and somatic/cybrid hybrids. Since mtDNA recombination is ubiquitous in distant hybridization, we put forward a hypothesis that the original CMS genes originated from mtDNA recombination during the germination of the hybrid seeds produced from distant hybridizations to solve the nucleo-cytoplasmic incompatibility resulting from the allogenic nuclear genome during seed germination.

## 1. Introduction

Cytoplasmic male sterility (CMS), defined as a natural phenomenon where plants fail in producing functional anthers, pollen, or male gametes, has been found in more than 150 species of plants [1]. CMS genes, located on mitochondria, have been extensively used for the production of hybrid seeds in various crops on a commercial scale due to the elimination of artificial emasculation. Since the maize (*Zea mays* L.) T-type CMS gene *orf13* was first discovered in 1986 [2], the rice (*Oryza sativa* L.) BoroII (BT)-type CMS gene *orf79* was found by Iwabuchi et al. (1993) [3], and the rice wild abortive (WA)-type CMS gene *orf352* was verified by Luo et al. (2013) [4]. Currently, at least thirty-two CMS related genes have been identified from thirteen crops, such as rice, maize, wheat (*Triticum aestivum* L.), oilseed rape (*Brassica napus* L.), sunflower (*Helianthus annuus* L.), sorghum (*Sorghum bicolor* (L.) Moench), and cotton (*Gossypium hirsutum* L.) [5]. Recently, in a novel polyploid rapeseed (*Brassica napus* L.), the CMS gene *orf188* for the enrichment of seed oil content was reported by Liu et al. (2019) [6]. Additionally, the Tadukan-type CMS gene *orf312* and Fujian abortive-type (FA-type) CMS gene *orf182* were found in rice [7,8]. All the reported CMS genes have been identified as chimeric structures in the mitochondrial DNA (mtDNA). The verification and mechanism of CMS genes have been studied continuously for more than 30 years.

Generally, hybrid seed production depends on a CMS gene in the mitochondrial genome (mitogenome) and a fertility restorer gene in the nuclear genome. Additionally, the interaction between the CMS gene and fertility restorer gene is also a good system to explore nuclear–cytoplasmic association [9]. The exclusive maternal inheritance of the mitogenome in most angiosperms can explain why CMS genes can be maintained from generation to generation through the hybridization between the sterile line and maintainer line (Figure 1). The maternal inheritance results in the preservation of conservative mtDNA sequences in angiosperm from generation to generation [10]. Due to the strictly maternal inheritance of mtDNA, when, where, and how mtDNA recombination and rearrangement occur deserve in-depth investigation.

Mitochondria have been speculated to originate from *α*-Proteobacteria in an endosymbiotic event over 1.5 billion years ago with residual genomes from ancient eubacteria [11,12,13]. Compared with the plant chloroplast genome and animal and fungal mitogenome, the plant mitogenome has undergone numerous changes in genome architecture, including frequent recombination, gene loss, gene transfer to the nuclear genome, foreign sequence capture, and genome structural variation, yet the coding sequences are highly conserved [14,15]. The behavior and significance of mtDNA recombination have not been fully understood, except for the perception that nearly all the reported CMS genes are chimeric structures resulting from mtDNA recombination [9,16].

As chimeric mitochondrial genes, CMS genes have been widely used and intensely studied. The questions of when and how the mtDNA recombination of maternal inheritance generates CMS genes still remain unsolved mysteries [17]. If the CMS problem is regarded as a tree, the application of a CMS gene is the crown, the action principle is the trunk, while the origin of the CMS gene is the root [18]. Despite the relative clearness of their application and action principle, the origin of CMS genes is still not clear. Therefore, we analyzed the common characteristics of CMS genes and reviewed the revealing processes of the original CMS genes, as well as the biological research progress of mitochondria and their DNA in recent years, which indicated that the occurrence of the original CMS genes is related to a distant hybridization. Plant mitochondria can be transmitted to the next generation through egg cells. New mitochondria do not arise de novo but from preexisting organelles such as promitochondria. However, mitochondria and the mitogenome are pleomorphic and dynamic in the plant life cycle. During the imbibition stage of seed germination, the special state of mitochondria and mtDNA with a wide range of crest structures provides the possibility for the recombination of the mitochondrial gene. Despite the stability of the mitogenome controlled by nuclear genes, mitochondria are regarded as semi-autonomous organelles regarding the mitogenome.

The mtDNA and mitogenome recombination has been widely found in the progeny of distant hybrids [19,20,21] and the contact zone between two hybridizing conifers [22]. The recombination and introgression of mtDNA during hybrid speciation have also been detected in yeast [23,24], insects such as Hymenopterans [25], and fish [26]. Therefore, we put forward a hypothesis that the mtDNA recombination during the germination of distant hybrid seeds is the primary way to acquire novel natural CMS genes. The heterogeneity of nuclear DNA (nDNA) in distant hybrid seeds may go against its interaction with mitochondria [21,27,28]. However, the absence of the relevant nuclear genome promoting good interactions with the mitogenome leads to mtDNA recombination, which has been considered as a solution mechanism for nucleo-cytoplasmic incompatibility [24,29]. For the origin of CMS genes and the mechanism of mtDNA recombination, this paper would not only advance a more objective and comprehensive understanding of mitochondria and their genome behavior but also enrich the CMS resources in the future. 

## 2. Structure and Origin of CMS Genes

### 2.1. Chimeric Configuration of Mitochondrial CMS-Associated Genes

The appearance of the CMS phenotype is the result of an interaction between a CMS gene and a nuclear fertility restoring (*Rf*) gene. Strong evidence [9,16] revealed that CMS genes result from mtDNA recombination. The sequencing of the entire mitogenome and sequence alignment between the sterile lines and fertile lines in various species [30,31,32,33,34,35] indicate that those mitochondrial genes with chimeric configuration are derived from mtDNA recombination. As shown in Table 1, most of these mitochondrial CMS-associated genes (MCAGs) consist of endogenous gene fragments and some chimeric sequences of unclear origin. 

The original CMS sources in crops found or created in the 1960s and 1970s are still utilized in the production of hybrid seeds currently due to their highly conservative mitogenome sequences and the corresponding maternal inheritance pattern. For this reason, the arising of chimeric mitochondrial genes such as CMS genes becomes so mysterious. For instance, how were CMS genes produced by the maternally inherited mtDNA in so many different plants? Despite the existence of other inheritance forms, maternal inheritance is still the main pattern of mtDNA transmission [66,67,68,69]. The maternal inheritance of mtDNA ensures that CMS genes can be transmitted from generation to generation by crossing between the CMS line and maintainer line (Figure 1). Therefore, when and how the mtDNA chimerism arose and led to the consistent CMS effect is an enigma, which could be explored through tracing the original CMS sources that have been reported. 

### 2.2. Original CMS Source from a Distant Hybridization

Despite many CMS lines being used in the hybrid seed production of various crops, the original CMS sources are not known. According to the relevant literature, the original CMS sources used in agricultural production can be obtained by three main approaches, including spontaneous appearance in natural population, distant hybridization, and somatic hybridization (i.e., symmetric somatic hybrids, asymmetric somatic hybrids, and cytoplasmic hybrids). Spontaneously sterile plants were found in wild relatives or cultivars, such as rice *WA-CMS*, which has been widely transferred into many sterile lines in different countries [70]. The polima-type cytoplasmic male sterility (*Pol-CMS*) was first identified in a Polish rape cultivar [71], while the *Ogura*-CMS of the radish *Ogura* sterile plant was originally identified in an unknown variety of Japanese radish [72]. 

Distant hybridization has been regarded as the most common way to create original CMS sources in various plants. For instance, the Honglian-type cytoplasmic male sterility (*HL-CMS*) [40], Boro II-type cytoplasmic male sterility (*BT-CMS*) [73], FA-type cytoplasmic male sterility (*FA-CMS*) [8], and dwarf wild-abortive cytoplasmic male sterility *(DA-CMS*) in rice [74], as well as the sorghum *IS1112C-CMS* (A3) [75], sunflower *Pet* [76], stem mustard *220-CMS* [53], pepper *PI164835-CMS* from India [77], and rapeseed *Cam-CMS* [78], all originate from distance hybridization. 

Somatic hybridization can also be original CMS sources, such as that revealed by the intergeneric somatic hybrids via asymmetric fusion between *Brassica napus* and *Arabidopsis thaliana* [79]. Asymmetric hybridization has been used to introduce existing CMS sources to another species and even genera, such as the rapeseeds *Ogura-CMS*, *Mori-CMS*, radish *Kos-CMS*, and petunia *Pcf-CMS* [80]. Their parents used in somatic hybridization usually belong to different species and even genera. Regardless of the different approaches, the generation of MCAGs essentially leads to the original CMS source (Table 1). 

The original CMS source can also be obtained by transgenic means through a nuclear gene, or through a mitochondrial gene. For instance, the expression of the nuclear gene *Msh1* is regulated by transgenic RNA interference (RNAi) to produce repeatable mtDNA recombination, which generates heritable CMS in tobacco (*Nicotiana tabacum* L.) and tomato (*Lycopersicon esculentum* Mill.) [81]. Another example is the transfer of the bean CMS gene *orf239* into tobacco to produce heritable CMS in tobacco [82]. However, these three original CMS sources have not yet been used for the production of hybrid seeds.

### 2.3. Close Connection between Original CMS Source and Distant Hybridization

Through comparison, most of the original CMS sources are obtained by distant hybridization (Table 1). The hybridization between different species, subspecies, and ecotypes within the same species or different genera, all termed distant hybridization, is a common method to create CMS in plants. A series of CMS lines have been developed through distant hybridization between similar wild and cultured plants by the International Institute of Rice [38]. In their parents used to create primitive male sterile plants in distant hybridizations, the fertility is almost normal. The rice *BT-CMS* has been created by the distant hybridization between Chinsurash Boro II (*Oryza sativa* L. subsp. *indica*, originated in India) and Taizhung 65 (a cultivated variety belonging to *Oryza sativa* L. subsp. *japonica*) (Shinjyo, 1969) [43], and the chimeric gene termed *orf79* leads to the CMS phenotype [39,83]. The rice *HL-CMS* caused by the chimeric gene *orfH79* originates from the distant hybridization between a wild rice (*Oryza rufipogon* Griff.) with red awn and growing in Hainan Province) and the cultivar Liantangzao (*Oryza sativa* L. subsp. *indica*) [70,84]. The *FA-CMS* and *D1-CMS* caused by the chimeric gene *orf182* originate from the distant hybridization between various *O. sativa* subsp. *indica* [8,42]. 

More than sixty types of rice CMS sources have been revealed to have originated from the distant hybridization in China [70]. Not only in the past are there several successful examples of CMS genes produced by distant hybridization in rice but also in the present. Recently, two CMS lines, i.e., Kalashree A and Padmini A, with a Miz. 21 cytoplasm have been developed through distant hybridization between *O. sativa* subsp. *indica* (from China) and *O. sativa* subsp. *indica* (from Indica), followed by repeated back-crossing with their respective recurrent male parents (Kalashree and Padmini) up to the BC_6_ generation [85]. Ten rice Dian-type CMS sources including their materials and procedures (Appendix A) have been revealed and recorded in detail by Yunnan Agricultural University. Recently, twelve new rice CMS sources have been created through distant hybridization between *O. sativa* subsp. *indica* and *O. sativa* subsp. *japonica*. These materials provided cytoplasm for various rice varieties such as Miyang 23, Diantun 502, IR58025 (IR58025/Nan29-48//Ce64), Yunhui290, and Minghui63. Their agronomic characters and fertility tend to be stable and contain several types of pollen abortion, including uninucleate, binucleate, and trinucleate pollen-stage abortion (Appendix A). 

As a crop that makes full use of heterosis through hybrid seeds, multiple CMS sources have been discovered and applied in sunflower. The sunflower *Pet-CMS* revealed by the distant hybridization between *Helianthus annuus* L. and *Helianthus petiolaris* L. is caused by the chimeric gene *orfH522* [76]. Thirteen original CMS sources have been created by inter-specific hybridizations, subsequently self-pollinated or consecutively back-crossed using six annual and nineteen perennial species of genus *Helianthus* as well as fourteen varieties and twelve inbred sunflower lines during the period of 1984–1996 [86]. Among the seventy-two original CMS sources in sunflower, most have originated from inter-specific hybridization [87]. Due to the importance of hybrid breeding, the production of new CMS sources has continued. Many CMS sources have been produced either by different inter-specific crossings involving *Helianthus argophyllus* (ARG-1), *Helianthus neglectus* (NEG-1), *Helianthus exilis* (EXI-2), and *Helianthus anomalus* (ANO-1) or intra-specific crossing between two subspecies of *Helianthus praecox* (i.e., PRR-1 and PRH-1) from 1974–2018 [19,88]. Vulpe (1972) [89] found a new CMS source, RIG-1, from the inter-specific crossing between a wild perennial hexaploid *Helianthus rigidus* and the cultivated *H. annuus*. Whelan (1980) [90] reported a new CMS source originated from the inter-specific crosses combining the nucleus of *H. annuus* cv. Saturn and the cytoplasm of *H. petiolaris.* Serieys and Vincourt (1987) [91] found many CMS sources by the crossing of wild annual species with cultivated sunflower. Christov et al. (2003) [88] reported plentiful new CMS sources (e.g., ARG-1, ARG-2, ARG-3, RIG-2, PET-4, PRH-1, PRR-1, and DEB-1) by inter-specific hybridization using different wild species including *H. argophyllus, H. petiolaris, H. rigidus, H. praecox*, and *Helianthus debilis*. Finally, *CMS PET2* originated from an inter-specific hybridization between *H. petiolaris* and *H. annuus*, which is totally different from *CMS PET1* in mechanism [19].

The sorghum *IS1112C-CMS*, created by the International Institute of Sorghum, has the Durra-Bicolor (originated from India) cytoplasm, wherein the chimeric gene involves *atp6* and *orf107* (ICRISAT, 1982) [75]. Inter-specific or intra-specific hybridization contributed to about 80% of the eighty-four original corn CMS sources found in America [92]. In oilseed rape, the stem mustard *220-CMS*-type was created by the distant hybridization between *Brassica campestris* and *Brassica juncea* and caused by the chimeric gene *orf220* [53]. A new CMS source, ICP 2039A, was produced by the distant hybridization between *Cajanus cajanifolius* and *Cajanus cajan* [93]. One of the CMS lines in soybean resulted from an inter-specific crossing between a wild-type Chinese soybean (*Glycine max* (L.) Merr.) and a wild annual soybean (*Glycine soja* Sieb.et Zucc) [94]. The new CMS source 1258A was produced by the distant inter-generic hybridization between XinJiang (*Sinapis arvensis* L.) as the female parent and Xiangyou15 (*Brassica napus* L.) as the male parent [95]. 

There are spontaneous cases of sterile cytoplasm proposed to have arisen from distant hybridization, such as the rice *WA*-CMS. When breeders at the Hunan Academy of Agricultural Sciences, China, performed two hybridizations in 1971, an original sterile plant was crossed to a nonglutinous (*Oryza sativa* L. var. *glutinosa*) variety in one crossing, while the original sterile plant was crossed to Nipponbare (*O. sativa* L. subsp. *japonica*) in another crossing. As proof that the *WA-CMS* originated from these crossings, some characteristics in the F_1_ plants, such as the plant type, spikelet type, and color of the tip of the lemma and kernel, can be traced to either of the parents. Furthermore, meiosis of pollen mother cells showed abnormalities in the derived *WA-CMS* plant [96]. 

Bonavent et al. (1989) [97] proposed that *Owen* cytoplasm of sugar beet (*Beta vulgaris* L.) originated from a crossing involving in the variety “Crapaudine”. CMS in *Petunia* Juss. is the result of a purposeful breeding program, wherein inter-specific hybrids are produced based on the successful generation of male sterility in tobacco [98,99,100]. Csillery et al. (1983) [101] and Woong et al. (1990) [102] obtained the sterile cytoplasm *1953 87-1* (*CMS-164813*) in pepper spontaneously from the distant hybridization between *Capsicum frutescens* L. and *Capsicum annuum* L. [103,104]. A plausible interpretation regarding how this may happen in nature is that the hybrids are produced from the distant hybridization between two co-exiting but distantly related species or genera because most of the CMS sources have a hybrid character [105]. 

Usually, original CMS source plants created by distant hybridization or found spontaneously have a high degree of fertility; only partial pollen is sterile. After more than 5–6 generations of back-crossing, the degree of sterility is usually improved [70].

## 3. Complexity of Angiosperm Mitochondria and Mitogenome

### 3.1. Variation in Mitogenome Size in Higher Plants

The plant mitogenome size ranges over 100-fold, from 66 kb in *Viscum scurruloideum* Barlow to 11,000 kb in *Silene conica* L. The increase in size has not resulted in an increase in gene number, and the number of mitochondrial genes in angiosperms is between 50 and 60 [10]. The animal mitogenome falls between 10 and 20 kb in size, while the fungal mitogenome falls between 30 and 90 kb [106]. By contrast, the plant mitogenome size varies widely within a same genus, even within the same species. Chang et al. (2011) [107] demonstrated a 1.6-fold variation in the mitogenome size among six mitotypes in *Brassica*. Particularly, Allen et al. (2007) [30] found that the mitogenome size in maize varied from 535,825 bp to 739,719 bp with large duplications (0.5–120 kb), accounting for most of the difference in the mitogenome. In sorghum, the mitogenome size ranged from 395,604 bp to 444,835 bp among seven accessions [108].

The mitochondria number is usually larger than the copy number of the mitogenome, and the copy numbers of mitochondria and mitogenome vary from organ to organ in the somatic cells of the same plant. The copy number of mitochondrial genes also changes in the same mitogenome. For instance, the copy numbers of mitochondrial genes such as *atp1*, *rps4*, *nad6*, and *cox1* not only differ from each other but also vary between different organs and show significant changes during the development of cotyledons and leaves in *Arabidopsis thaliana* [109]. Post-natal maturation of the oocytes for ovulation requires further mtDNA replication activity, which causes a burst in mtDNA copy number [110,111]. Over the plant life cycle, the lowest mitochondrial copy number exists in dry seeds, two per cell. Additionally, the mitochondria are in a promitochondrial state without complete functional morphology in dry seeds [112].

### 3.2. Complex Mitogenome Structure and Its Dynamic Change

Generally, the plant mitogenome is often depicted as a circular ‘master chromosome’ comprising all mitochondrial genes and non-coding sequences, which exhibits wide variations among different species. Although a single circular structure has been assembled by using the mitogenome of *Arabidopsis thaliana* [106,113], the mitogenome of *Silene conica* (sand catchfly, Caryophyllaceae) is arranged into numerous circular chromosomes [114,115,116]. In *Broussonetia* spp. (Moraceae), the mitogenomes of *Broussonetia monoica* and *Broussonetia papyrifera* consist of a single circular structure, whereas the *Broussonetia kaempferi* mitogenome has a double circular structure. Remarkably, except for a few transfer RNA (tRNA) genes, their gene content was same [117]. Three circular-mapping molecules (lengths 312.5, 283, and 186 kb) were assembled from the *Populus simonii* mitogenome, and all had protein-coding capabilities [118]. However, the cucumber (*Cucumis sativus*) mitogenome was assembled into one large circular chromosome (1556 kb) and two small circular chromosomes (45 and 84 kb), of which only the large chromosome has protein-coding capability [119]. The copy number of the large chromosome is approximately twice as abundant as the two small chromosomes, indicating the independent replication of the three mitochondrial chromosomes in cucumber plant cells [106,119]. Wang et al. (2019) identified extensive whole-genome rearrangements among the kiwifruit mitogenomes and found a highly variable V region in which fragmentation and frequent mosaic loss of intergenic sequences occurred, resulting in greatly inter-specific variations [120]. Additionally, similar to recombination intermediates, the linear, highly branched, and occasionally circular molecules with tails are found in soybean (*Glycine max* (Linn.) Merr.) [121]. Most plant mitogenomes can be mapped as single “master circles”, which maintain a dynamic equilibrium as a set of sub-genomes [80]. Structurally, plant mtDNA occurs as circular, linear, and branched subgenomic molecules of various sizes in the mitogenome [122,123]. Kozik et al. (2019) [124] found that the simple, circular molecules cannot accurately describe the true nature of plant mitogenome, and confirmed that plant mitogenomes are a complex, dynamic mixture of forms based on long-read and short-read sequencing data of three *Lactuca* species.

Mitogenomes show rapid change along with mitochondria fusion and fission during seed germination [112,125]. Cheng et al. (2017) [112] observed physical structures of DNA molecules isolated from mitochondria at the early stage during seed germination of mung bean (*Vigna radiata*). The mtDNA becomes more branched and longer from simple linear molecules shorter than its genome size, which gradually acquire a size longer than the genome during seed stratification, imbibition, and germination. Then, the mtDNA reverts to simple short linear molecules after seed germination.

An ongoing transfer of genes to the nuclear and chloroplast genomes primarily causes the variation in the plant mitogenome regarding size and gene content, which disrupts gene continuity in introns or exons of flowering plants [12]. For example, one of the duplicated plastid *rps13* genes originating from the *A. thaliana* nuclear genome has been recruited to encode a mitochondrially targeted polypeptide [126]. A sequence widespread in the carrot mitogenome transferred to the plastid genome of a carrot ancestor is integrated into a tRNA promoter of the plastid *trnV* gene, which replaces the original promoter sequence [127]. In cashew (*Anacardium occidentale* L.), a ~6.7-kb fragment of mtDNA is found to be inserted into the plastome IR [128]. In a survey about mitochondrial genes of angiosperms, *rpl*, *rps*, and *sdh* genes were confirmed to be the easiest to be lost from the mitogenome in some angiosperm lineages [129,130].

### 3.3. Variation in mtDNA Sequences

The variation in the mtDNA is mainly caused by point mutations, deletions, or aberrant recombinations. In contrast to little plant mtDNA variation resulting from base substitutions [131], most angiosperms’ mtDNA variation aspects have been shown to result from rearrangements. Generally, plant mtDNA recombination can be divided into two types: (1) gain-of-function, such as the acquisition or expression of a CMS-causing chimeric ORF; and (2) loss-of-function, such as the alteration of essential gene [132]. A gap similar to the deletion of short sequences can often be found after mtDNA recombination of several different haplotype sequences.

### 3.4. Conserved Sequences and Recombination Characteristics

The organizational complexity of plant mitogenomes reflects a propensity for mtDNA to rearrange, which results from a high level of recombination across repeated sequences [30,133]. Frequent and reversible recombination between pairs of relatively large (>1 kb) repeats can form alternative molecules of these dynamic genomes. Due to the function that mitochondria provide energy for individual and coding sequences alteration usually leading to death, and because the known mitochondrial genes are highly conserved in angiosperm species [134,135,136], a low substitution rate of about 1.65–7.04 substitutions per 10 kb in pairwise comparisons is found in most angiosperm mtDNA [137].

### 3.5. Substoichiometric Shifting 

Due to many mitogenomes per cell, a new mtDNA mutation or recombination usually does not lead to detectable alteration of the phenotype, until enough variations have accumulated. Thus, all mtDNA mutations (or recombinations) are expected to exist at a very low (“substoichiometric”) level [132]. A phenomenon that renders the plant mitogenome unusually variable in structure is termed substoichiometric shifting (SSS) [138]. Generally, dramatic and rapid changes in mtDNA stoichiometries often accompany ectopic recombination [64,139]. The asymmetric DNA exchange at small repeats influences the stoichiometry of subgenomic mtDNA molecules [140]. Rampant structural rearrangements via repeat-mediated recombinations make for extensive but heritable structural variation within individuals, which can also be defined as SSS [141]. No matter which definition of SSS, we should first consider the timing of the mtDNA reorganization, and then consider the copy increase in subgenomic mtDNA molecules.

During seed germination, promitochondria develops into mature mitochondria, which is accompanied by mitochondria fusion and fission, and the most drastic change in mtDNA. Generally, the copy increase in normal mitochondria is in the stage of mitochondria fission. If subgenomic mtDNA molecules still exist in the cell of mature tissue, we need to ask why the copy number has no increase in the stage of mitochondria fission. We have to understand mitochondria fusion and fission, which contribute to the comprehension of SSS and mtDNA behavior. However, the behaviors of mitochondria and mtDNA still remain mysterious. To improve our comprehension of SSS, we need to answer the following questions: (1) what mitochondrial fusion and fission mean for the variation in mtDNA; and (2) whether a variation mechanism similar to non-sister chromatid exchange and recombination during nuclear DNA meiosis exists during seed germination. 

## 4. Mitochondria and mtDNA Behavior during Seed Germination

### 4.1. Promitochondria in Dry Seed and Mitochondrial Biosynthesis during Seed Germination

The reason why the CMS phenotype always occurs in distant hybrid offspring is still unknown. After all, CMS lines are completely preserved through the crossing with maintainer lines. Recently, gradually increasing studies have promoted the understanding of the inheritance and evolution of mtDNA. Mitochondria show state changes at various stages in the plant life cycle [112,125,142,143]. A state of suspended animation involving promitochondria, structurally and functionally deficient relative to typical mitochondria extracted from mature tissue and lacking defined cristae and internal structure, is found in dry seeds [143,144,145]. The promitochondria have been regarded as specialized transgenerational genetic vaults that could re-differentiate during germination for autotrophic growth [142]. Both promitochondria and plastids are small and lesser in number at the early stage of germination, while they are both more numerous and larger towards the end of germination, with a characteristic morphology [144]. Along with seed imbibition of maize (*Zea mays* L.), rice (*Oryza sativa* L.), and Arabidopsis (*Arabidopsis thalian*), promitochondria developed into more typical mature mitochondria within 12–24 h at 4 °C, also accompanied by increasing metabolic activity [94,114]. This transition from hollow promitochondria in dormant seeds to fully functional mitochondria with a wide range of crest structures and various biochemical activities requires many proteins [146]. Mitochondria contain their own genomes that encode about fifty proteins, and various ribosome RNA (rRNA) and transfer RNA (tRNA) in plants. However, the remaining over 1000 proteins located in mitochondria are encoded by nuclear genes, which are translated in the cytosol and introduced into mitochondria [147,148,149]. The proteins encoded in mitochondria exist in multiple subunit complexes and are also composed of nuclear coding subunits. Giraud et al. (2010) [150] revealed that site II elements bind various plant-specific families of transcription factors involved in the regulation of a variety of nuclear genes encoding mitochondrial proteins. Most mitochondrial proteins must be imported into the mitochondria from the cytosol, indicating that the import of mitochondrial protein is a vital part of mitochondrial biogenesis. The rate of protein import into mitochondria might be determined by the inner membrane translocases [148].

### 4.2. Mitochondrial Fusion and Fission during Seed Germination

Followed by mitochondrial biogenesis, mitochondria exhibit random and localized oscillatory movements. The mitochondrial movement is gradually accelerated alongside seed imbibition, which brings about an increase in the interaction and fusion of mitochondria. Mitochondria at higher movement speeds continue to be more organized up to the stage of testa rupture. At this stage, the dramatic reorganization of chondriome (refers to all mitochondria in a cell, collectively) involves massive mitochondrial fusion and fission, and membrane biogenesis to form a perinuclear tubuloreticular structure, enabling the mixing of previously discrete mtDNA nucleoids. At the end of seed germination (i.e., endosperm rupture stage), along with the reduction in mitochondrial movement speed, the number of reticular and tubular mitochondria has decreased for the rising of oval-shaped mitochondria values [142]. Additionally, the growing proportion of small mitochondria is concomitant with the increase in total number because of mitochondrial fission [151,152]. 

The mtDNA content and its complex structure change progressively with cotyledon development during seed germination of mung bean (*Vigna radiata* L.). After 2 h seed imbibition at 4 °C, their linear fragments of mtDNA are converted into a simple rosette structure in cotyledon cells. After 12 h, nearly all mtDNA has been converted to a longer linear form, which is accompanied by the simultaneous disappearance of the rosette core. The rosette core, consisting of condensed mtDNA and a replication starting sequence, plays an initial and central role in recombination-dependent replication [112]. These phenomena indicate that, because of mitochondrial movement during seed germination, the mixed mtDNA coupled with its rosette structure provide better opportunities for intramolecular recombination. The active processes of massive mitochondrial fission and fusion require multifarious specialized proteins, such as mechanical enzymes that physically alter mitochondrial membranes, and adaptor proteins that regulate the interaction of mechanical proteins with organelles [153,154,155].

The massive mitochondrial fusion and fission take place in the organs with active cell division, such as stem apical meristems, root apical meristem, and regenerating protoplasts [142,156,157]. The visualization of mitochondria in living tissues confirms the high interaction within the mitochondrial population, i.e., their highly mobile cytoskeletons regarding movement with no uniform or static entities within the cell, ensuring the mutual contact of physically discrete organelles [143]. Mitochondria can change their number, composition, and function in different cells, organs, and species through frequent fusion and fission to respond to various stresses [158]. Different from the massive mitochondrial fusion in non-plant systems, the detectable nucleoids with a proportion over 67% are distributed evenly throughout the population following massive mitochondrial fusion and fission in seed germination [151], which indicates that the frequent and transient massive mitochondrial fusion and fission help to overcome the heterogeneity of mtDNA. The massive mitochondrial fusion and fission, playing an important role in the life cycle of flowering plants, might promote the transmission to the nucleus, mtDNA recombination, and homogenization of mitochondrial components [157]. 

### 4.3. Timing of mtDNA Recombination

#### 4.3.1. MtDNA Recombination in Plants

The physical structure of mtDNA is now thought to be mostly linear molecules of various lengths, with random fragments from a long head-to-tail concatemer sequence of the mitogenome, some even longer than the genome size [159,160,161,162]. At the seed imbibition stage, the massive mitochondrial fusion and fission bring their DNA molecules together, which provides an opportunity for the recombination of mtDNA fragments. Then, the quantity and quality of mtDNA as well as its recombination can be tightly and specifically controlled [142]. In regenerated cytoplasmic hybrids (i.e., cybrids), the unique mtDNA band distinguished from both parent mtDNA and from a mixture of the two, having been identified after the fusion of two different cells, which confirmed the occurrence of mtDNA recombination between mitogenomes from their parents [163]. Sanchez-Puerta et al. (2015) [116] found more than thirty-five recombination regions between homologous sequences in the F_1_ mitogenome after cytoplasmic fusion between two species of *Solanaceae* (i.e., *Nicotiana tabacum* and *Hyoscyamus niger*). An illegitimate recombination caused by a microhomology-mediated break-induced replication was found in the mtDNA of the repeat cybrid between tobacco (*Nicotiana tabacum*) and henbane (*Hyoscyamus niger*) [164]. Previous studies have revealed that plant somatic hybridization often results in the recombination of the mitogenome [165,166,167]. 

#### 4.3.2. The mtDNA Recombination during Distant Hybridization in Yeasts and Animals

In addition to plants, inter-specific mtDNA recombination has been widely reported in a broad range of organisms including yeast hybrids [168], reef building corals [169], salmonid [170,171] and cyprinid fishes [172,173], and primates [174]. In *Saccharomyces*, the diverse and highly reticulated mtDNA showed signatures of recombination within or between species [175,176,177,178]. Based on mitogenome sequence, Leducq et al. (2017) [23] found that both parental species contributed to the hybrid mitogenome through mtDNA recombination. The frequent mtDNA recombination between parental types in experimental crosses recreates the early step of speciation events. Similar to the nuclear genome, the mitogenome can, therefore, also play a role in hybrid speciation. The analysis of recombined mtDNA sequences in six hybrids of *Saccharomyces cerevisiae* × *Saccharomyces paradoxus* by Illumina MiSeq and restriction fragment length polymorphisms (RFLPs) revealed that the rearranged molecules were composed of a major skeleton from one parental molecule and small regions substituted with segments from another parental molecule [168]. Extensive genotype-specific mtDNA recombination events were observed in a collection of 864 hybrid yeast lines. However, the rate of mtDNA recombination cannot be predicted by parental divergence. Hybridization in yeast induced mtDNA degeneration demonstrated a profound impact on mtDNA evolution, metabolism, and the emergence of reproductive isolation between species [24]. 

Part sequencing of the mitogenome (8141 bp) in twenty-eight specimens of *Hucho taimen* from six localities in the Amur River basin provided a detailed architecture of recombination events due to anthropogenic hybridization between salmonid fishes such as Siberian taimen (*Hucho taimen*) and two lenok subspecies (i.e., *Brachymystax lenok* and *Brachymystax lenok tsinlingensis*) [171]. Balakirev (2022) [26] analyzed the patterns of nucleotide diversity in the complete mitogenome of icefishes including *Pampus chinensis, Neosalanx tangkahkeii*, and other closely related salangid species and detected clear signals of mtDNA recombination. The hybridizations might lead to mtDNA changes, which contributed to the subsequent generation of new lineages in goldfish-like fish [179]. The paternal mtDNA could be transmitted into the mtDNA of homodiploid lineages, thus demonstrating its significance in inherited mtDNA. Additionally, Ujvari et al. (2007) [180] provided evidence for mtDNA recombination between two mitochondrial haplotypes in the hybrid zone in the Australian frillneck lizard (*Chlamydosaurus kingii*) through the sequencing of the entire mitogenome.

## 5. Interaction between Mitochondrial and Nuclear Genomes

### 5.1. Stability and Quantity of Mitogenome Controlled by Nuclear Loci

The maintenance and expression of the mitogenome as well as its stability and quantity are controlled by the nucleus or nuclear loci. For instance, the *Msh1* gene, located in the *Arabidopsis* nuclear genome, involves the suppression of mtDNA rearrangements during development, particularly the illegitimate recombination of mtDNA in plants, due to the encoded proteins being targets to both mitochondria and plastids. Generally, the expression of the *Msh1* gene maintains the stability of the mitogenome [181]. The inhibition of *Msh1* gene expression in tobacco and tomato by RNAi leads to the rearrangement of the reproduceable mtDNA and the production of heritable CMS genes [81]. Because the *CHM* gene actively suppresses the copy number of the subgenomic molecule carrying the chimeric mitochondrial sequence, its loss results in the rapid and specific copy number amplification of the subgenomic molecule, which produces the consequent leaf variegation [138]. 

The pollen fertility of CMS common bean (*Phaseolus vulgaris* L.) is controlled and restored by a single dominant nuclear fertility restorer gene designated *Fr*. This restoration is accompanied by mtDNA rearrangements in the restored plants, such as the loss of an independently replicated subgenomic DNA molecule in the mitogenome [182]. The model of autonomous replication of plant subgenomic mtDNA molecules is supported and observed in maize with stoichiometric changes in different molecules in response to different nuclear backgrounds [183]. 

Previous studies have confirmed that the stability of mtDNA is controlled by nuclear genes. The gene *RECX* with the interaction of gene *Msh1* has been extensively studied due to its involvement in the suppression of ectopic and illegitimate recombination in both organellar compartments. Except for the energy supplied by functioning mitochondria, the communication between mitochondria and other cellular processes have surpassed the exchange or transport of known metabolites, which is accompanied by sequential and dynamic gene expression, protein synthesis, and post-translational modifications during seed germination [154]. However, their complex regulatory and communication network have not been revealed until now (Figure 2), particularly how two or more different nuclear genomes coordinate with the cytoplasm genome coming from one of the parents in diploid or allopolyploid cells [184]. The mtDNA recombination in seed germination reduces the heterogeneity between the nuclear genome and mitogenome, which explains the similar issue in distant hybrid offspring from two heterologous genomes coordinating with the cytoplasm genome coming from one of the parents.

### 5.2. Contradiction between Heterologous Genome in Nuclei and Mitogenome

Mitochondrial function is part of a complete cellular network, and the communication between mitochondria and other cellular processes goes beyond the exchange or transport of known metabolites [154]. For instance, a small number of subunits making up complexes in crucial metabolic processes are encoded by the mtDNA, while a vast majority are encoded by the nuclear genome and transported to the mitochondria. Opportunities for mitonuclear incompatibility, the mismatch between nuclear and mitochondria-encoded components, can thus arise not only in the subunits of complexes that must work together but also in the recognition and importation of necessary proteins [185]. To complicate matters further, mitochondria are not restricted solely to metabolic processes. They also play important roles in a wide range of other processes such as immunity and apoptosis, which require just as many mitonuclear interactions [186,187,188]. Similar to distant hybridization, one nuclear genome is heterologous to mtDNA from another source genome in the cybrids, which may be an obstacle to their mutual recognition. The development of alloplasmic lines demonstrates a close connection with the structural and functional variability of nuclear and organelle genomes [189,190]. 

## 6. Mechanism of the Origin of CMS Genes

### 6.1. CMS Genes Originate from mtDNA Recombination during Hybrid Seed Germination

The dynamic equilibrium could be maintained through the continuous communication and a high level of compatibility between the mitochondrial and nuclear genomes, which have been confirmed in the above examples and discussion. To achieve the co-existence of mtDNA and nDNA from different sources in the same system of completing life activities, massive mitochondrial fusion and fission occur because of the mitochondrial movement at high speed and then are accompanied by the mtDNA recombination. Regarding the CMS phenotypes frequently appearing in the offspring of distant hybrids, we propose a hypothesis that natural CMS genes originate from the mtDNA recombination through the massive mitochondrial fusion and fission during the germination stage of distant hybrid seeds to overcome the communication barriers between heterologous nuclear genomes and the mitogenome (Figure 3). The maternal mitochondria in the cytoplasm may be one of the inducements for the co-existence of two heterologous nuclear genomes in the same nucleus. For instance, the nDNA in heterozygotes have higher recombination and mutation rates than heterozygotes [191]. The mtDNA recombination could be considered an adaptive mechanism to the foreign nuclear genome. The deletion of the corresponding restoring genes in the nuclear genome contributes to the expression of the CMS gene and appearance of the CMS phenotype. Therefore, the CMS genes can be transmitted generation to generation by the crossing between the sterile line and maintainer line due to the absence of nDNA heterogeneity and nucleo-plasmic interaction; there is no need for mtDNA recombination.

### 6.2. Molecular Mechanism and Sequence Characters of mtDNA Recombination

In general, duplicative horizontal gene transfer (HGT) and differential gene conversion might contribute to the generation of functional, transcompartmental chimeric mitochondrial genes from the mtDNA chimera, which usually originate from the anciently related sequences in chloroplast DNA and mtDNA [192,193]. In the repeated cybrid between tobacco and henbane, the break-induced replication (BIR) pathway is thought to be the main mechanism responsible for mtDNA recombination [164]. Twenty-eight homologous recombination events involve three main mechanisms, including the BIR pathway, double Holliday Junction (DHJ) pathway, and synthesis-dependent strand annealing (SDSA) pathway [164]. Plant mitochondria have been confirmed with enzymatic activities for mtDNA recombination [121]. The short repeat sequences (<100 bp in length) in the dynamic mtDNA rearrangements result in the creation of chimeric mitochondrial genes [183,194,195], which have been proved to be responsible for the chimeric mitochondrial genes, such as CMS genes in radish (*Raphanus sativus* L.) and sugar beet (*Beta vulgaris* L.) [35,190]. The computational reconciliation analysis of the 1685-kb mitogenome of cucumber (*Cucumis sativus*) revealed that the proliferation of the dispersed repeats by the recombination events, expansions of the existing introns, and the acquisition of sequences from diverse sources, such as the cucumber nuclear and chloroplast genomes, explain the large size of the cucumber mitogenome [119]. Additionally, as a locus for *indica–japonica* hybrid male sterility, *Sa*, located on chromosome 1 in hybrids between *O. sativa* subsp. *indica* and *O. sativa* subsp. *japonica*, comprises two adjacent genes, *SaF* and *SaM*, which are diverged by nucleotide variations in wild rice (*Oryza rufipogon* Griff.) [196]. The mitogenome sequence analysis revealed the frequent loss and acquisition of mtDNA from nuclear or plastid genomes, and a ubiquitous presence of repeat sequences in higher plants [10,14,15,107,119]. Eighteen different haplotypes of the *B-atp6-orfH79* sequence were found in wild rice (*Oryza rufipogon* Griff.), suggesting that mtDNA recombination results from transposable elements. We systematically analyzed the MCAGs to determine the consistent conserved domain database (CDD) [197]. The results confirmed that no consistent CDD was found in all the MCAGs (Table 1); i.e., no fixed mitochondrial genes were involved in the chimeric structure of the sterile genes. Therefore, the generation of CMS genes overcomes the communication barriers between the two heterologous genomes in the hybrid seed from distant hybridization through the mtDNA recombination during germination.

## 7. Conclusions

We summarized the sequence characteristics of the CMS genes that have been reported in plants and confirmed that all CMS genes belong to chimeric mitochondrial genes. CMS genes can be maintained from generation to generation through the hybridization between the CMS sterile line and maintainer line, which utilizes the faithfully maternal inheritance of mtDNA in production practice. However, when and how so many chimeric genes are produced in the highly conservative mtDNA become problems for phytology to explain. 

The plant mitogenome has the characteristics of sequence conservation, large variation in size, and a complex structure, which maintains a dynamic equilibrium over the life cycle. The frequent sequence recombination forms chimeric structures, which leads to geometric multiple changes in the size of the mitogenome but with a limited number of about 50–60.

We traced back many original CMS sources that have been reported in various crops and found that distant hybridization was the primary source to generate an original CMS.

The mitochondria cannot be synthesized de novo. Numerous studies show different forms and structures of mitochondria and mtDNA during various stages in the life cycle of angiosperm. The mtDNA structure changes accordingly along with the mitochondria form, such as the promitochondrion in seeds, mitochondrial biosynthesis, and massive mitochondrial fusion and fission during seed germination, thus providing the conditions for mtDNA recombination.

Indeed, mtDNA recombination is often found in the offspring of distant hybrids, and cybridization occurs, in which their parents are different species and even different genera. Studies on animals and yeasts confirm the existence of mtDNA recombination in the offspring of distant hybridization with direct evidence.

In this review, we proposed that most CMS genes originate from the mtDNA recombination during the hybrid seed germination from distant hybridization. The mtDNA recombination may be an adaptive mechanism in cytoplasmic–nuclear interaction to overcome the communication barriers between two heterologous genomes in the nucleus. The mtDNA stability is dependent on nDNA. We hope this work will trigger further studies on mitochondria and mtDNA. 

## Figures and Tables

**Figure 1 ijms-25-08372-f001:**
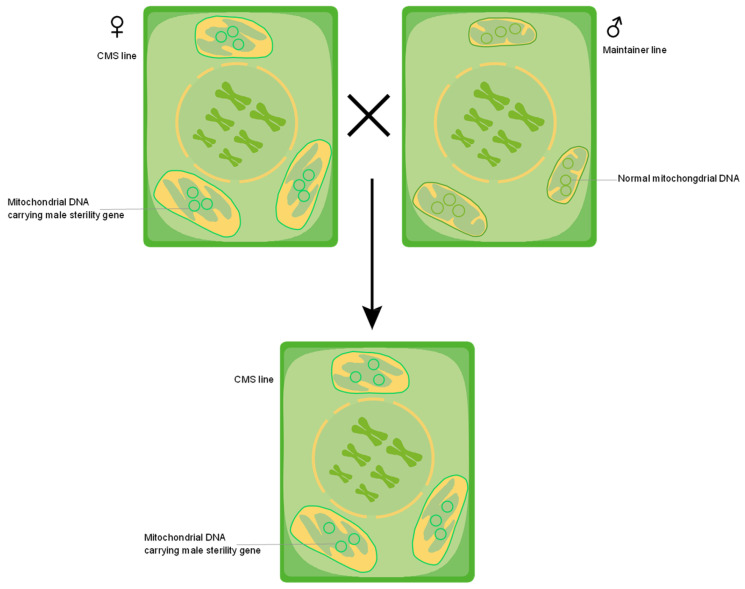
Cytoplasmic male sterility (CMS) gene was maintained by the hybridization between sterile line and maintainer line.

**Figure 2 ijms-25-08372-f002:**
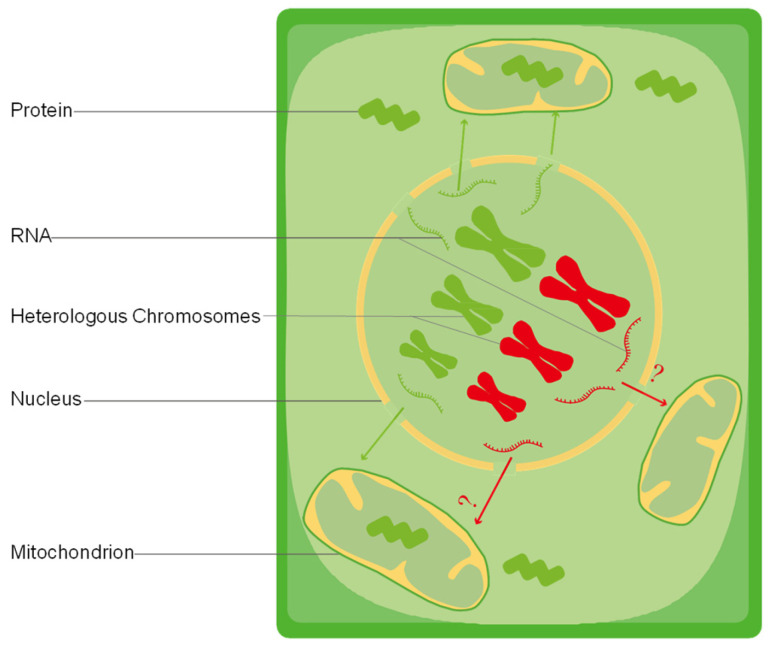
How two heterologous genomes coordinate with the cytoplasm genome coming from female parent becomes a question. Green indicates the genome came from the female parent; red indicates the genome came from the male parent. ? indicates structural proteins in mitochondria formerly encoded by the paternal genome that is replaced by heterogenous genome. If the offspring survive, how do nuclear and mitochondrial genomes overcome this barrier? How do mitochondria recognize each other after the genes for gene expression, protein synthesis, and transcription regulation are encoded by the paternal genome that is replaced by heterologous genomes?

**Figure 3 ijms-25-08372-f003:**
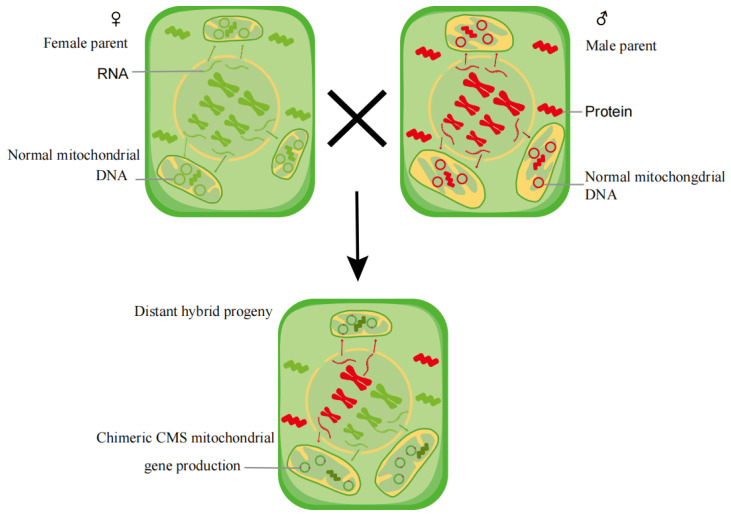
The generation of CMS gene in hybrid seed to overcome communication barriers between two heterologous genomes.

**Table 1 ijms-25-08372-t001:** Chimeric genes involved in cytoplasmic male sterility (CMS) and their corresponding sterile cytoplasms.

Material	Type	Chimeric Gene	Gene Involved	Origin Style	CDD	References
*Oryza sativa* L. subsp. *indica*	*WA*	*WA352*	*orf284*, *orf288*, *orf224*	Spontaneous	ATGACGAGAG	Luo et al., 2013 [4]
*Oryza sativa* L. subsp. *indica*	*WA*	*orfB*	*atp6*	Spontaneous	GGATAATCCG	Das et al., 2010 [36]
*Oryza sativa* L. subsp. *indica*	*WA*	*AY295770*	*rps-rp116-nad3-rps12*	Spontaneous	ACGGCCCTCA	Yashitola et al., 2004 [37]
*Oryza rufipogon* Griff	*CW*	*orf307*, *NC_013816*	*coxⅡ*, *orf288*, *orf224*	Distant hybridization	TGTTTATCAT	Fujii et al., 2010 [32]
*Oryza sativa* L. subsp. *indica*	Chinsurah *Boro II*	*orf79*, *D14339*	*atp6*	Distant hybridization	ATGGCAAATC	Akagi et al., 1994 [38]; Wang et al., 2006 [39]
*Oryza sativa* L. subsp. *indica*	*HL*	*orf 79*	*atp6*	Distant hybridization	ATGACAAATC	Hu et al., 2012 [40]
*Oryza sativa* L. subsp. *japonica*	*Lead*	*L-orf79*	*atp6*	Distant hybridization	ATGACAAATC	Kazama et al., 2016 [41]
*Oryza sativa* L. subsp. *indica*	*FA*	*FA182*	*-*	Distant hybridization	No CDD	Jiang et al., 2022 [8]
*Oryza sativa* L. subsp. *indica*	*D1-CMS*	*orf182*	*-*	Distant hybridization	No CDD	Xie et al., 2018 [42]
*Zea mays* L.	*T*	*urf13*	*26S-ribosomal*	Distant hybridization	CGTCAATGAT	Dewey et al., 1986 [2]
*Zea mays* L.	*S*	*orf355*/*orf77*	*atp9*, *orf221*	Distant hybridization	ATGGAAGATA ATGTTTGCAT	Zabala et al., 1997 [43]
*Sorghum bicolor* (L.) Moench	*IS1112C(A3)*	*atp6* *x57100*	*atp6*	Distant hybridization	GTTCGTGTTC	Kempken et al., 1991 [44]
*Sorghum bicolor* (L.) Moench	*IS1112C(A3)*	*orfF107*	*orf209*, *atp9*	Distant hybridization	ATGTCGCGAC	Pring et al., 1999 [45]
*Sorghum bicolor* (L.) Moench	*IS1112C(A3)*	*orf265/orf130*	*T-urf13*, *atp6-2*; *orf25*	Distant hybridization	ATGAACGGTC	Tang et al., 1996 [46]
*Triticum timopheevi* Zhuk	*T. timopheevi*	*orf256*	*cox1*	Distant hybridization	-	Hedgcoth et al., 2002 [47]
*Brassica napus* L.	*Pol*	*orf224*	*atp6*	Spontaneous	GGATGCTACT	Singh et al., 1991 [48]
*Brassica napus* L.	*Nap*	*orf222*	*nad5*, *orf139*	Protoplast fusion	ATTAATCTAA	L’Homme et al., 1997 [49]
*Brassica napus* L.	*Ogura*	*orf158*, *orf138*, *z12626*	*tRNA-fMet*	Cybrid induce	TGCCTCAACT ATGATTACCT	Bonhomme et al., 1992 [50]
*Brassica napus* L.	*Tour*	*orf193*,	*atp9*, *cob*, *nad2*	Cybrid induce	-	Dieterich et al., 2003 [51]
*Brassica juncea* L.	*Tour* (*juncea*)	*orf263*, *x83692*	*atp6-nad5*	Distant hybridization	ATGAAAAATA	Landgren et al., 1996 [52]
*Brassica juncea* L.	*220*-type	*orf220*, *AAO59387*	*atpA*	Distant hybridization	-	Yang et al., 2009 [53]
*Brassica juncea* L.	*Mori*	*orf108*, *EF483940*	*atpA*	Cybrid induce	ATGAATACTA	Ashutosh et al., 2008 [54]
*Brassica oleracea* L.	*Mur*	*orf72. AB243571*	*atp9-2*, *rps7*	-	ATGGAAATTC	Shinada et al., 2006 [55]
*Raphanus sativus* L.	*Ogura*	*atp6.* *M24672*	*orf105*, *fMet-tRNA*	Cybrid induce CrGC15	AGTGATACAT	Makaroff et al., 1989 [56]
*Raphanus satiuus* L. cv. *Kosena*	*Kos*	*orf125*	*orfB*	Spontaneous	ATGATTACCT	Iwabuchi et al., 1999 [57]
*Raphanus sativus* L.	*Ogura*	*orf138*	*orfB*	Spontaneous	ATGATTACCT	Krishnasamy et al., 1993 [58]
*Helianthus annuus* L.	*Pet*	*orfH522-atpA*, *x55963*	*cob*, *atp9*	Distant hybridization	ATGCCTCAAC	Köhler et al. 1991 [59]
*Helianthus annuus* L. ssp. *tetanus*	*Ant1*	*atp6* *X82386*	*atp6*, *atp9*, *atpA, nadl + 5 and coxIII*	Distant hybridization	ATGTGACTGA	Spassova et al. 1994 [60]
*Phaseolus vulgaris* Linn	*G08063*	*pvs-orf239*, *M87062*	*cob3*, *psb*	Spontaneous	ATGGATCATTTGTTCCTCCC	Janska et al., 1998 [61]; Abad et al., 1995 [62]
*Beta vulgaris* subsp. vulgaris	*Owen*	*atpa*, *X68691*, *atp6*, *X54722*	*atpA*, *atp6*	Spontaneous	ATGGAATTCTATGGGTACTC	Xue et al., 1994 [63]
*Capsicum annuum* Linn	*PI164835* from India	*ψatp6-2*, *DQ126680*	*atp6*	Spontaneous	ATGATGCGAC	Kim et al., 2006 [64]
*Petunia parodii*	*Pcf*	*orf402*	*atp9*, *cox2*, *urfs*	Cybrid induce	TCGTGATGGA	Young et al., 1987 [65]

-, There is no corresponding published DNA sequence available for analysis. CDD, conserved domain database.

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
