# Peer review of "A Systematic Review and Developmental Perspective on Origin of CMS Genes in Crops"

_ijms, 2024, doi:10.3390/ijms25158372_

Round 1

Reviewer 1 Report

Comments and Suggestions for Authors

Comments and remarks:

 The present work represent a review on the studies aimed to elucidate the mechanism of phenomenon of cytoplasm male sterility (CMS) in plants that causes fail in producing functional anthers, pollen or male gametes, and is applied for the production of hybrid seed. It tracing back reported original CMS sources confirmed that distant hybridization is the main approach to generate original CMS source in natural populations and breeding practice. The common characteristics of CMS genes was analyzed, and revealing process of original CMS genes, as well as biological research progress of mitochondria and its DNA in recent years were reviewed.

For the attention about origin of CMS genes and the mechanism of mtDNA  ecombination, this paper would not only advance a more objective and comprehensive  understanding of mitochondria and its genome behavior, but also enrich original CMS resources in the future.

In the present review, the requirements of the "International Journal of Molecular Sciences" for the compilation of this type of manuscripts have been met. The problem discussed is well presented with appropriate background, tables and diagrams. The statements made in the Conclusions are based on the exposition of previous studies carried out on the target problem.

 The following remarks ken be made:

-          The abstract can be shortened and an introductory sentence can be added, such as: "The present review presents the research on cytoplasmic male sterility (CMS) in plants to elucidate the mechanisms of this phenomenon"

In conclusion the manuscript is recommended to be published in “International Journal of Molecular Sciences”.

Author Response

The present work represent a review on the studies aimed to elucidate the mechanism of phenomenon of cytoplasm male sterility (CMS) in plants that causes fail in producing functional anthers, pollen or male gametes, and is applied for the production of hybrid seed. It tracing back reported original CMS sources confirmed that distant hybridization is the main approach to generate original CMS source in natural populations and breeding practice. The common characteristics of CMS genes was analyzed, and revealing process of original CMS genes, as well as biological research progress of mitochondria and its DNA in recent years were reviewed.

For the attention about origin of CMS genes and the mechanism of mtDNA ecombination, this paper would not only advance a more objective and comprehensive understanding of mitochondria and its genome behavior, but also enrich original CMS resources in the future.

In the present review, the requirements of the "International Journal of Molecular Sciences" for the compilation of this type of manuscripts have been met. The problem discussed is well presented with appropriate background, tables and diagrams. The statements made in the Conclusions are based on the exposition of previous studies carried out on the target problem.

Response:Thanks for your high recognition of our review. It is a great honor to us. Through tracing back the relevant literature about the origin of cytoplasmic male sterility (CMS) genes in various plants, we put forward a hypothesis that the original CMS genes originate from mtDNA recombination during the germination of hybrid seeds produced from the distant hybridization, to solve the nucleo-cytoplasmic incompatibility brought from the allogenic nuclear genome during seed germination. Therefore, we hope this review not only advance a more objective and comprehensive understanding of mitochondria and its genome behavior, but also enrich original CMS resources in the future.

The following remarks ken be made:

-The abstract can be shortened and an introductory sentence can be added, such as: "The present review presents the research on cytoplasmic male sterility (CMS) in plants to elucidate the mechanisms of this phenomenon"

In conclusion the manuscript is recommended to be published in “International Journal of Molecular Sciences”.

Respond: Thanks for your detailed comment about the retrenchment of abstract. The abstract in the new revision has been shortened from 372 words to 274 words. Several unimportant contents such as “As natural phenomenon that plants fail in producing functional anthers, pollen or male gametes”, “ However, mitochondria and mtGenome behavior are not yet fully understood”, “The former has been fully verified in the practice for its maintenance through the crossing of sterile line and the maintain line, while the later leads to geometric multiple change of mtGenome size with limited genes number”, “Spontaneous sterile plant usually has hybrid characteristics to frequently produce original CMS genes”, have been removed from the abstract.

Reviewer 2 Report

Comments and Suggestions for Authors

The manuscript provides a comprehensive overview of the structural and functional diversity of plant mitochondrial genomes (mitogenomes), highlighting their unique features compared to other eukaryotes. The authors have done an excellent job of synthesizing a large body of research and presenting it in a clear and well-organized manner. One of the strengths of the manuscript is the detailed coverage of the mechanisms underlying the maintenance of mitogenome stability, such as the involvement of the RECX and Msh1 genes. The authors also provide a good overview of the phenomenon of substoichiometric shifting (SSS), which renders plant mitogenomes unusually variable in structure. However, the authors could further expand and update Chapter 3, which covers the "Structural Diversity of Plant Mitochondrial Genomes." This section could be strengthened by incorporating more recent findings on the size, stability, and variation of mitogenomes at both the interspecific and intraspecific levels. Additionally, the title could be more focused on manuscript content, a review should rather summarize current knowledge in the field. Overall, the manuscript is a valuable contribution to the field of plant mitogenomics, and with the suggested improvements, it would be an excellent resource for researchers and students interested in plant mitogenomics.

Author Response

The manuscript provides a comprehensive overview of the structural and functional diversity of plant mitochondrial genomes (mitogenomes), highlighting their unique features compared to other eukaryotes. The authors have done an excellent job of synthesizing a large body of research and presenting it in a clear and well-organized manner. One of the strengths of the manuscript is the detailed coverage of the mechanisms underlying the maintenance of mitogenome stability, such as the involvement of the RECX and Msh1 genes. The authors also provide a good overview of the phenomenon of substoichiometric shifting (SSS), which renders plant mitogenomes unusually variable in structure. However, the authors could further expand and update Chapter 3, which covers the "Structural Diversity of Plant Mitochondrial Genomes." This section could be strengthened by incorporating more recent findings on the size, stability, and variation of mitogenomes at both the interspecific and intraspecific levels.

Respond: Thanks for your basic recognition and constructive suggestions about our paper. In the re-submission, we have enhanced the content about “Structural Diversity of Plant Mitochondrial Genomes” with about six new added references as follows: “In addition,  sorghum mitogenome size ranged from 395,604 bp to 444,835 bp among seven sorghum accessions” (in section 3.1.), and “In Broussonetia spp., the mitogenomes of Broussonetia monoica and Broussonetia papyrifera consisted of a single circular structure, whereas Broussonetia kaempferi mitogenome was uniquewith a double circular structure. Remarkably, except for a few transfer RNA (tRNA) genes, their gene contents were consistent. Three circular-mapping molecules (lengths 312.5, 283, and 186 kb) assembled from of Populus simonii mitogenome, all had protein-coding capability. However, the cucumber (Cucumis sativus) mitogenome was assembled into one large circular chromosome (1556 kb) and two small circular chromosomes (45 and 84 kb), of which only the large chromosome has proteincoding capability. The copy number of the large chromosome is approximately twice as abundant as the two small chromosomes, indicating the independent replication of the three mitochondrialchromosomes in cucumber plant cells. Wang et al. (2019) has identified extensive whole-genome rearrangements among kiwifruit mitogenomes, and found a highly variable V region in which fragmentation and frequent mosaic loss of intergenic sequences occurred, resulting ingreatly interspecific variations ” (in section 3.2).

Additionally, the title could be more focused on manuscript content, a review should rather summarize current knowledge in the field.

Respond: Thanks. The title has been corrected as “A Systematic Review and Developmental Perspective on the Origin of CMS Genes in Crops”, which could summarize the whole manuscript content.

Overall, the manuscript is a valuable contribution to the field of plant mitogenomics, and with the suggested improvements, it would be an excellent resource for researchers and students interested in plant mitogenomics.

Respond: Thank you for your high praise of our paper. We would continue the research in the field of plant mitogenome and CMS genes in crops with the hope to further reveal generation mechanism of CMS phenomenon in molecular science level.

Reviewer 3 Report

Comments and Suggestions for Authors

The manuscript "A Hypothesis: CMS Genes Originate from Mitochondrial DNA  Recombination during Germination of Distant Hybrid Seeds" is a review paper, but at the same time, authors formulate an hypothesis for the origin of CMS genes. Although, hypothesis are more than welcome, I fail to see a strong evidence for the one raised in this study. This might be also due to the fact that several sections are very hard to follow - many have grammatical mistakes or are incomplete, preventing any reader to follow the text. Text flow is also very hard to follow. For instance, the authors start to say "Cytoplasm male sterility (CMS), defined as natural phenomenon that plants fail in producing functional anthers, pollen or male gametes, has been found in more than 150 species of plants". Yet, latter, the authors then write "Until now, at least thirty-two CMS related genes have been identified from thirteen crops, such ..."- Thus, the remaining species where CMS have been found are wild ones? In this context, the text needs to be better organized, and sentences need to be revised.

In relation to the figures, how were they produced?

In relation to Table 1, what do the different colours indicate? How were those genes found? I

Comments on the Quality of English Language

See above.

Author Response

The manuscript "A Hypothesis: CMS Genes Originate from Mitochondrial DNA Recombination during Germination of Distant Hybrid Seeds" is a review paper, but at the same time, authors formulate an hypothesis for the origin of CMS genes.

Respond: Thanks for your specific comments of our paper. We track back the breeding and preservation of sterile line inproduction practice, and the published literature about the breeding of CMS line, the verification and sequence analysis of CMS gene, as well as mitochondrial and mtDNA behavior during whole plant life cycle, particularly during the seed germination. This review also summarize the state and behavior of mitochondrial and mtDNA behavior during different plant life cycle, such as the development and maturation of promitochondria along with mitochondrial fusion and fission as well as mtDNA recombination in seed germination. Based on the above foundations, we carefully provided the hypothesis that CMS genes might originate from mitochondrial DNA recombination during germination of distant hybrid seeds. The previous studies have laid a solid foundation and provided scientific evidences for this hypothesis.

The mtDNA recombination is indisputable fact. All known CMS genes are chimeric mitochondrial DNA. In the production practice, the sterile line can be preserved through hybridization between sterile line and maintainer line, which should be attributed to the matrilineal inheritance of mtDNA. When and where occur mtDNA recombination in so conservative matrilineal inheritance of mtDNA, and why. Until now, there is no definite answers to these questions. Indeed, no one pays attention to these issues.

In this review, we answered the above key issue as follows. Where? The mtDNA recombination largely and frequently occur in distant hybrids and somatic/cybrid hybrids. When? The mtDNA recombination occur at the imbibition stage during the seed germination due to abundance mitochondrial fusion and fission as well as the formation of mtDNA rosette structure at this stage. Why? The objective of mtDNA recombination in distant hybrids and somatic/cybrid hybrids would be to solve the nucleo-cytoplasmic incompatibility brought from the allogenic nuclear genome during seed germination.

Although, hypothesis are more than welcome, I fail to see a strong evidence for the one raised in this study.

Respond: Thanks. Above evidences and more detailed discussions in the review support this hypothesis. Until now, there was no more rational hypothesis about formation and origin of CMS genes, even no one put forward these questions. However, to avoid controversy, the title has been corrected as “A Systematic Review and Developmental Perspective on the Origin of CMS Genes in Crops”, which reconcile the retrospect about relevant research progress and new perspective about the origin of CMS genes in crops.

This might be also due to the fact that several sections are very hard to follow - many have grammatical mistakes or are incomplete, preventing any reader to follow the text. Text flow is also very hard to follow. For instance, the authors start to say "Cytoplasm male sterility (CMS), defined as natural phenomenon that plants fail in producing functional anthers, pollen or male gametes, has been found in more than 150 species of plants". Yet, latter, the authors then write "Until now, at least thirty-two CMS related genes have been identified from thirteen crops, such ..."- Thus, the remaining species where CMS have been found are wild ones?

Respond: Thanks. Cytoplasm male sterility (CMS) is a natural phenomenon that plants fail in producing functional anthers, pollen or male gametes, due to the existence of relevant CMS genes in crops. The artificial distant hybridization also could generate CMS phenomenon, which indicated that the wild CMS phenomenon originate from the distant hybridization. No matter natural sterile plants or artificial  sterile plants, both could be used for production of sterile line. Although CMS phenomenon have been found in over 150 plants, not all CMS genes have been identified in plants. The research about CMS gene focused on the major crops for the seed production. Therefore, only about 32 CMS genes have been identified.

In this context, the text needs to be better organized, and sentences need to be revised.

Respond: Thanks for your comments. This paper has been carefully revised, such as Abstract and 3. Complexity of angiosperm mitochondria and mitogenome. Additionally, a native English scholar assist English polishing to reduce grammar and spelling errors. If necessary, we hope this paper could be polished with the assistance from professional company to enhance readability.

In relation to the figures, how were they produced?

Respond: Thanks. The figures were produced by the “Adobe Illustrate”.

In relation to Table 1, what do the different colours indicate? How were those genes found?

Respond:In order to distinguish CMS sources, we used different color to mark. Specifically, the sterile source from natural sterile plants was marked using blue color as spontaneous origin style. The sterile source from distant hybridization was marked green color. The sterile source from somatic hybridization was marked water blue. The discovery process of related CMS genes can refer to cited literature.

Round 2

Reviewer 3 Report

Comments and Suggestions for Authors

I have no further suggestions to this version.